# Optical detection of single sub-15 nm objects using elastic scattering strong coupling

MohammadReza Aghdaee [1], Melissa J. Goodwin[2] & Oluwafemi S. Ojambati [1] ✉

Metallic nano-objects play crucial roles in diverse fields, including biomedical imaging, nanomedicine, spectroscopy, and photocatalysis. Nano-objects smaller than 15 nm exhibit extremely low scattering cross-sections, posing a significant challenge for optical detection. An approach to enhance optical detection is to exploit nonlinearity of strong coupling regime, especially for elastic scattering, which is universal to all objects. However, there is still no observation of the strong coupling of elastic light scattering from nano-objects. Here, we demonstrate the strong coupling of elastic light scattering in self-assembled plasmonic nanocavities formed between a gold nanoprobe and a gold film. We employ this technique to detect individual objects with diameters down to 1.8 nm. The resonant mode of the nano-object in the nano-cavity environment strongly couples with the nanocavity mode, revealing anti-crossing scattering modes under dark-field spectroscopy. The experimental result agrees with numerical calculations, which we use to extend this technique to other metals. Furthermore, our results show that scattering cross-section ratio of the nano-object scales with the electric field to fourth power, similar to surface-enhanced Raman spectroscopy. This work establishes a new possibility of elastic strong coupling and demonstrates its applicability for observing small, non-fluorescent, Raman inactive sub-15 nm objects, complementary to existing microscopes.

Nano-scale objects are ubiquitous in physical, chemical, and biological systems[1]. Small metallic nanoparticles, in particular, are widely applied in various fields, including catalysis[2], spectroscopy[3,4] and sensing[5], biomedicine for drug delivery[6], in biomedical imaging for contrast enhancement[7,8], and electronics to create transistors and logic circuits[9]. Metallic nanoparticles also have detrimental effects as environmental pollutants[10], defects in integrated circuits[11], and are possibly bio-toxic, resulting in cancer[12]. It is, therefore, of broad interest to investigate and characterize nanoparticles.

Despite their importance, it is challenging to optically detect and image nanoparticles that are much smaller than the wavelength of visible light. This challenge is due to the scattering cross-section that scales $\sigma_{\text{sca}} \propto R^6$, where $R$ is the radius of the nanoparticle, following Mie theory[13]. The nonlinear scaling implies that the scattered intensity by a

10 nm nanoparticle is $10^6$ times less than a 100 nm nanoparticle. As a result, far-field optical techniques can directly detect the elastic light scattering by individual larger nanoparticles (-100 nm)[14]. However, a single smaller object (with diameter < 15 nm, which we refer to as "nano-object" henceforth) scatters light so weakly that observation using a standard optical microscope is challenging. In addition, many nano-objects do not have fluorescence and phosphorescence due to the absence of a radiative electronic transition in the visible region. Moreover, the nano-objects do not have distinct vibrational modes that can be detected using Raman or infrared spectroscopy. The weak elastic light scattering and extinction are the only optical signals from these nano-objects.

Existing optical techniques use different approaches to obtain signals from the nano-objects[15]. Fluorescence-based techniques

[1]Faculty of Science and Technology, MESA+ Institute for Nanotechnology, University of Twente, Enschede, The Netherlands. [2]Nanolab, MESA+ Institute of Nanotechnology, University of Twente, Enschede, The Netherlands. ✉e-mail: o.s.ojambati@utwente.nl

require fluorescent markers, which photobleach and have a limited photon count[16–18]. Scattering-based techniques avoid these limitations and increase scattering contrast using interference[19–21], a scanning probe to collect nearfield[22], and using intensity modulation[23]. The drawback of these techniques is mainly the low signal-to-noise ratio of the measured scattered light of small objects, and/or they have a slow scan speed. Another approach is photothermal microscopy, where a heated nano-object changes the refractive index of an embedding medium while a probe beam monitors the change[24,25]. However, the technique requires an embedding environment with a high photo-thermal response, which is impractical in many cases. In addition, a nano-object can cause a spectral shift in the resonance wavelength of a nanophotonic structure, providing a signature of the nano-object. Examples of such structures include a nanorod[26], nanowires[27], photonic crystals[28], ring resonators[29], and photonic waveguides[30]. However, the sensitivity is extremely low, using nanophotonic structures; For example, there is a < 0.01 nm wavelength change for a 1 nm change in diameter of nano-object[31–35]. The low sensitivity also inhibits the detection of sub-15 nm objects.

In this work, we introduce a completely different approach to boost the detection sensitivity and the scattered intensity of nano-objects. We consider the nano-object as a resonator that can strongly couple to a plasmonic nanocavity. The coupling between the elastic scattered intensity of the two systems needs to exceed both the cavity loss rate and the scattering loss rate of the nano-object for energy exchange between the nano-object and nanocavity[36]. If the strong coupling condition is fulfilled, we expect that two scattering modes will appear, and their spectral shift will be highly dependent on the properties of the nano-object. Even though this approach utilizes the hybridization of plasmonic modes, it is still unexplored for the detection of a single nano-object experimentally[37].

Strong coupling has been widely observed in several systems where light and matter are hybridized and new dressed states arise. Typically, strong coupling involves an optical cavity and either an electronic (vibronic) state (e.g., in an atom[38,39], quantum dot[40–42], two-dimensional material[43,44], or single molecule[45–48]) or a molecular vibrational state[49–51]. On the other hand, there is no demonstration of elastic scattering strong coupling to detect small nano-objects, even though elastic scattering is universal to all objects due to refractive index contrast.

This paper presents an experimental and numerical demonstration of elastic scattering strong coupling to detect single sub-15 nm nano-object. The nano-object strongly couples with a plasmonic nanocavity, which is formed by a Au nanoparticle placed on top a Au film (Fig. 1a). The strongly coupled system reveals two modes in dark-field extinction spectroscopy that strongly depend on the cavity and nano-object parameters. We demonstrate that this effect applies to metallic nano-objects such as gold, silver, aluminum, and copper. Employing elastic light scattering strong coupling, establishes a new possibility of observing small, non-fluorescent, Raman inactive objects.

## Results
### Modes of a Au nano-object inside a plasmonic nanocavity
We perform numerical calculations to understand the interactions of a single nano-object and a plasmonic cavity by solving the full Maxwell equations using the boundary element method[52–54] and finite element method (for numerical setup, see Methods). A broadband light source with transverse magnetic polarization illuminates the structure at an angle of 65° (Fig. 1a). The diameter $d$ of the nano-object varies from 1 nm to 15 nm, and the diameter of the nanoprobe is 80 nm. We account for non-local effects that are dominant at smaller sizes (<20 nm) of the nano-objects[55] using two different models: In hydro-dynamic model, a composite object consisting of the nano-object and a thin layer coating with effective permittivity describes the non-local effects[56–58]. In a model based on Feibelman parameters, a mesoscopic boundary condition is applied to include the non-local effects[59] (see Methods).

Our starting point is the plasmon resonance of the nano-object outside the nanocavity environment. The non-local hydrodynamic model shows a pronounced resonance shift of up to 180 meV as the diameter of the nano-object decreases from 15 nm to 1 nm (Fig. 1b). The result of using the Feibelman parameters agrees well with the non-local hydrodynamic model. In contrast, a local (Drude) model shows no pronounced spectral shift. To verify these numerical results, we experimentally probe the plasmon resonance of colloidal suspension

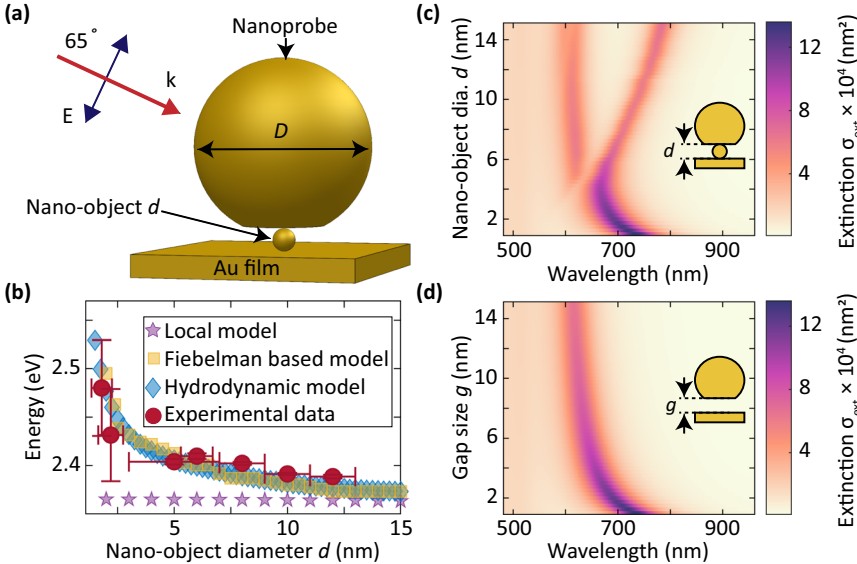

**Fig. 1 | Light scattering of sub-15 nm objects inside and outside a plasmonic nanocavity. a** Schematic of the sample that consists of a faceted Au nanoparticle (nanoprobe) with diameter $D$ and a nano-object with diameter $d$ that is placed on top of the Au film. The incident angle of light is 65°. **b** Plasmon resonance of sub-15 nm Au nano-objects in water colloidal suspension, outside the nanocavity. **c** Extinction cross-section ($\sigma_{ext}$) spectra of a Au nano-object inside nanocavity for different nano-object diameters. **d** Extinction cross-section ($\sigma_{ext}$) spectra of an empty nanocavity with different gap size $g$. In (**c**, **d**), the Au nanoprobe diameter $D = 80$ nm and $g = d$.

of Au nano-objects in water and observe that the plasmon resonance from the non-local model and Feibelman-based model agrees with the experimental results. These results align with the well-established understanding that as the nanoparticle diameter decreases below 20 nm, the plasmon resonance shifts towards higher energies[60]. The spectral shift is due to spatial dispersion of the dielectric function $\varepsilon(\mathbf{r}, \mathbf{r}')$, and this means that the dielectric function contributes to the displacement field at positions $\mathbf{r}$, which is non-local to the position $\mathbf{r}'$ of the excitation electric field[58,61,62].

We calculate the scattering spectrum of the nano-object inside the nanocavity using the hydrodynamic model due to its accuracy and fast computation speed to investigate a wide range of sample parameters space. There is a remarkable difference between the extinction spectra when a nano-object is in the gap compared to an empty nanocavity (Fig. 1c, d). For transverse magnetic excitation, two plasmonic modes appear for nano-objects with diameters $d > 4$ nm, and only one mode is visible for $d < 4$ nm. These modes are also present in the scattering and absorption cross-section spectra (Supplementary Information, SI. A). The longer wavelength mode strongly depends on $d$, while the lower wavelength mode closely follows the mode of the empty nanocavity. We establish the origin of these modes in the following section below. For transverse electric excitation, only a single mode exists, and it does not depend on the diameter of the nano-object (SI. B). For the empty nanocavity, a dipolar mode due to the coupling between the

nanoprobe and the image charge persists for all gap spacing[37,63]. We henceforth refer to this dominant scattering mode of the empty cavity as the nanocavity mode.

The nano-object-in-nanocavity modes show a nonlinear wavelength dependence on the diameter of the nano-object, resulting in a high sensitivity. We define the sensitivity as $\frac{\partial \lambda_m}{\partial d}$ (where $\lambda_m$ is the mode wavelength), and we found a sensitivity from 12 to 70. This implies that there is a maximum wavelength change of 70 nm for a 1 nm change of $d$ (for more details, see SI. C). This is a remarkable sensitivity which is about three orders of magnitude higher than existing techniques[31-35].

### Experimental probe of a single nano-object in the nanocavity

To experimentally confirm the modes described above, we fabricate plasmonic nanocavities with a nano-object inside the gap. The nanoprobe assembles directly on the nano-object on a Au film using electrostatic attraction (Methods). The surfaces of the nanoprobe and nano-object are oppositely charged due to the capping agents of citrate and cysteamine hydrochloride, respectively (Fig. 2a). The surface functionalization with cysteamine hydrochloride does not perturb the physical properties of the nano-object (Methods). We first deposited several 12 nm nano-objects on the Au film (inset, Fig. 2b). The surface density of the nano-objects is aimed to be 1600 particles/$\mu m^2$, to obtain an average particle spacing of 25 nm. This surface density ensures that on average only one nano-object is

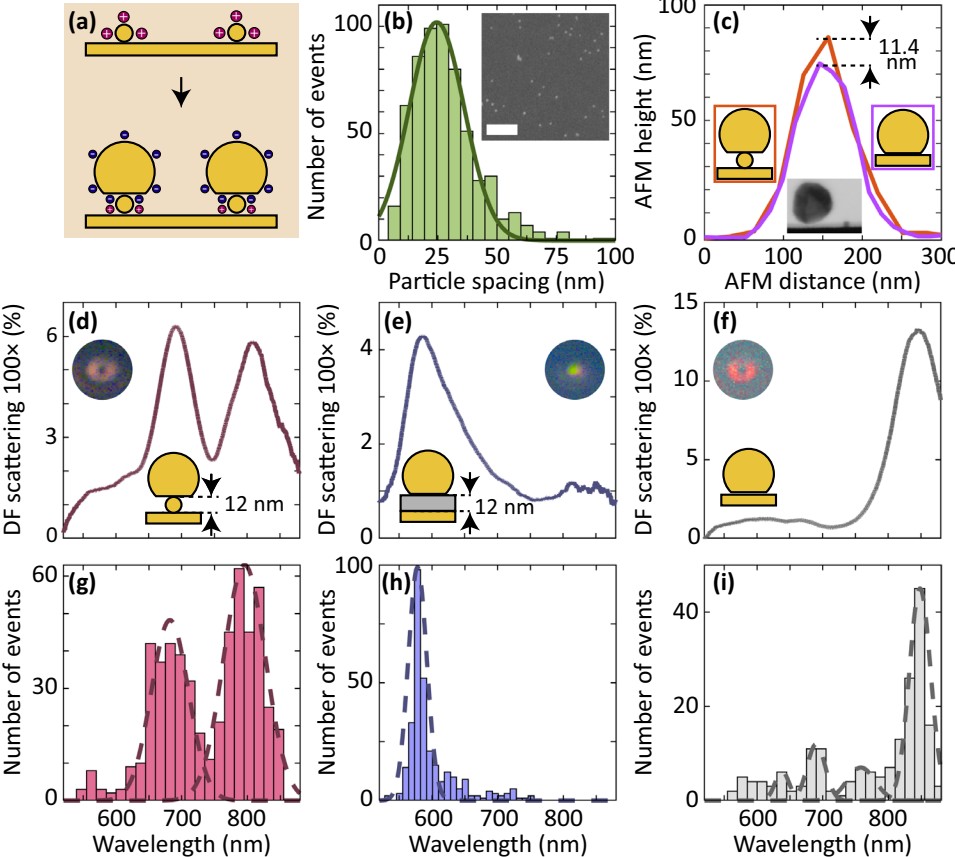

**Fig. 2 | Experimental signature of a 12 nm nano-object inside a nanocavity.** **a** Schematic of the sample fabrication process, which includes depositing positively charged nano-objects on the Au film (top) and forming the nanocavity by using electrostatic attraction of the nano-object and a negatively charged nanoprobe (bottom). **b** The spacing distribution between the nano-objects determined from the scanning electron microscope image in the inset. The scale bar of the inset is 200 nm. **c** The AFM height measurement of an empty nanocavity (purple line) and a nanocavity with a 12 nm nano-object inside the gap (orange line). Inset: A scanning electron microscope cross-sectional image of a nano-object beside a probe with a nano-object embedded underneath. The dark-field spectra of a nanocavity with (**d**) a 12 nm nano-object inside the gap, (**e**) a dielectric layer filled the gap, (**f**) nano-particle directly on the Au film. In (**d**–**f**), the insets show the dark-field image of the corresponding nanocavity. **g**–**i** The histogram of the scattering peak measured nanocavities for (**g**) a 12 nm nano-object inside the gap, (**h**) a dielectric layer filled the gap, (**i**) an empty gap. The dashed line is a Gaussian fit to the detected maxima in the histogram. In (**a**–**i**), the nanoprobe diameter is 80 nm.

under the probe facet[64]. Based on the SEM image analysis, the nano-object spacing with the highest occurrence frequency is 24.5 nm, as expected (Fig. 2b). After depositing the nanoprobe, we confirm that the nano-object is inside the nanocavity, using an atomic force microscope (AFM) to measure the height of the nanocavities on two different samples: (1) the probe on top of the nano-object (12 nm in diameter) on the Au film and (2) the probe directly on the Au film. There is an $11.4 \pm 3.3$ nm height difference between the maximum height of a nanocavity with the 12-nm object inside the gap and an empty nanocavity (Fig. 2c, and SI. D for more statistics). We also use focused ion beam (FIB) milling to make a cross-section around a single compound nano-object inside the nanocavity. Thereafter, SEM imaging of the sample cut-out enables a visualization of the content of the nanocavity. The resulting SEM image reveals that the nanoprobe is positioned at a height similar to the diameter of a nearby nano-object (Fig. 2c inset and for more images, see SI. D). However, attempts to further FIB mill the sample cut-out, the compound structure disassembles (see more discussion in SI. D). Together, the FIB-SEM and AFM data confirm that the nano-object is inside the nanocavity.

We use a custom-made dark-field micro-spectrometer to acquire scattering images and the extinction spectrum to probe the resonance of the compound structure (for experimental setup, see SI. E). For a single nano-object ($d = 12$ nm) inside the nanocavity, there are two dominant resonant modes at 690 nm and 807 nm (Fig. 2d). This experimental observation is consistent with the simulation results that also reveal two modes. To further prove that these modes are due to the Au nano-object in the gap, we compare the modes to that of two other samples: (1) a nanocavity with a dielectric layer (aluminum oxide) with thickness 12 nm, which is equal to $d$ and (2) an empty nanocavity that has the Au probe directly on the Au film. The nano-object-in-nanocavity modes are significantly different from the modes of these two samples (Fig. 2e, f). There is a single plasmonic mode in the dark-field spectrum of nanocavity with a dielectric-filled gap and empty gap at 585 nm and 844 nm, respectively. These modes are the dominant nanocavity modes with different gap sizes. The dielectric-filled nanocavity has a blue-shifted resonance compared to the empty nanocavity because of the larger gap size.

Far-field scattered intensity profiles provide additional information about the nature of the modes of the nano-object-in-the-cavity structure[65]. The nano-objects alone are not visible in the dark-field image due to the low scattering cross-section. On the other hand, the nano-object-nanocavity compound structure scatters light significantly, such that there is a high contrast between the structure and the Au film (insets, Fig. 2d–f). The scattering image from the nano-object inside the nanocavity and the empty nanocavity is a ring-shaped donut pattern. This far-field scattering pattern is due to the electric field that is z-polarized inside the nanocavity, and the z-oriented dipole dominates the far-field emission[66,67]. However, the dielectric-filled gap nanocavity shows a spot pattern due to the transversal plasmonic mode of the nanoparticle.

We further measured the dark-field scattering spectrum from more than 700 nanocavities with a nano-object inside the gap, dielectric-filled gap, and empty gap (Fig. 2g–i). For each structure, the peak wavelength of the resonance modes was determined using a Gaussian distribution fit to the histogram maxima. There are two dominant peaks at 682 nm and 796 nm for 12 nm nano-objects inside the nanocavity, while the dielectric-filled gap nanocavity and empty nanocavity have a single dominant peak at 578 nm and 848 nm, respectively. The histograms provide sufficient statistics to demonstrate the consistency of the modes.

## Size dependence of the nano-object-in-cavity mode

We investigate the dependence of the nano-object-in-cavity modes on the nano-object diameter from 1.8 nm to 12 nm and interrogated

>250 structures for each diameter. The resonance modes of 8 nm and 10 nm nano-objects in the nanocavity show two strong peaks at (655, 712) nm and (644, 758) nm, respectively, while the 2.2 nm nano-objects show one strong peak at 680 nm (Fig. 3a–c). A peak with a lower occurrence frequency appears at around 800 nm for 2.2 nm and 8 nm nano-objects. We attribute this peak to nanocavities that have more than one nano-object (Supplementary Fig. 8). There is a broader distribution of the resonance peaks for nanocavities with a nano-object compared to empty and dielectric-filled nanocavities. We attribute this broadening to the large variation in $d$ of the nano-objects, as supplied by the manufacturer (Methods).

We extract the peak wavelength of all the scattering modes from the histograms for different nano-object diameters. There are two resonance modes for nano-objects from 12 nm down to 6 nm, while the smaller nano-objects show one dominant mode (Fig. 3d). The higher wavelength mode shows a stronger dependence on $d$ than the lower wavelength mode. The measured resonances agree well with the simulation results (yellow line) extracted from Fig. 1c. The relative occurrence frequency of the mode in each histogram determines the data marker size. The vertical error bar is determined from the standard deviation of the Gaussian distribution fit to the histograms, and the horizontal error bar is from the distribution of the diameter of the nano-object obtained from analyzing the SEM images (SI. D). The lower wavelength mode slightly deviates by about 15 nm from the simulation results, which might be due to experimental parameters (e.g., the polyhedral shape of the probe, specific vertical arrangement of the probe, etc) that are not accounted for in the simulations (SI. G, H). To a good extent, the experimentally observed modes are reproduced in the simulation.

We define the sensitivity of our experimental results as the change in resonance wavelength with respect to the diameter of the nano-objects. Experimentally, we observe a maximum sensitivity of $35 \pm 9$. In contrast, the simulation results show a higher maximum sensitivity of 71 (SI. C). This difference arises because the simulation considers a finer tuning of the diameters of the nano-object and nanoprobe, which therefore enabling us to find the optimal parameter combinations that gives the highest sensitivity. Although this detailed parameter search is tedious experimentally, the sensitivity measured is of a similar order of magnitude to the simulation result. The experimentally obtained sensitivity is approximately three orders of magnitude higher than that of existing techniques (SI. C).

## Strong coupling of nano-object and plasmonic nanocavity

To understand the origin of the nano-object-in-cavity mode, we describe the interaction of the nano-object in the plasmonic nanocavity using a simple model of two coupled oscillators. The oscillators are the nano-object on the Au film and the nanocavity formed by the probe close to the Au film. Coupling between resonators can be categorized as weak, moderate, or strong based on the relation between coupling strength and system losses. Weak coupling leads to enhanced emission without spectral splitting, while moderate coupling shows partial mode hybridization and spectral asymmetry such as Fano resonance. In strong coupling, the coherent energy exchange between the nano-object and the nanocavity dominates losses, resulting in Rabi splitting[68]. We simulate the extinction of these two systems separately and tune the resonance of the nanocavity by changing the probe size. The energy $E^{\pm}$ of the resonance of the strongly coupled systems can be described using[68,69]

$$E^{\pm} = \frac{E_{nc} + E_{no}}{2} \pm \frac{1}{2} \left[ (\hbar\Omega_R)^2 + (E_{nc} - E_{no})^2 \right]^{1/2}, \qquad (1)$$

where $E_{nc}$ and $E_{no}$ are the energy of the nanocavity mode and the nano-object in the nanocavity environment, respectively. The $E_{no}$ also includes perturbation of the resonance of the nano-object due to an

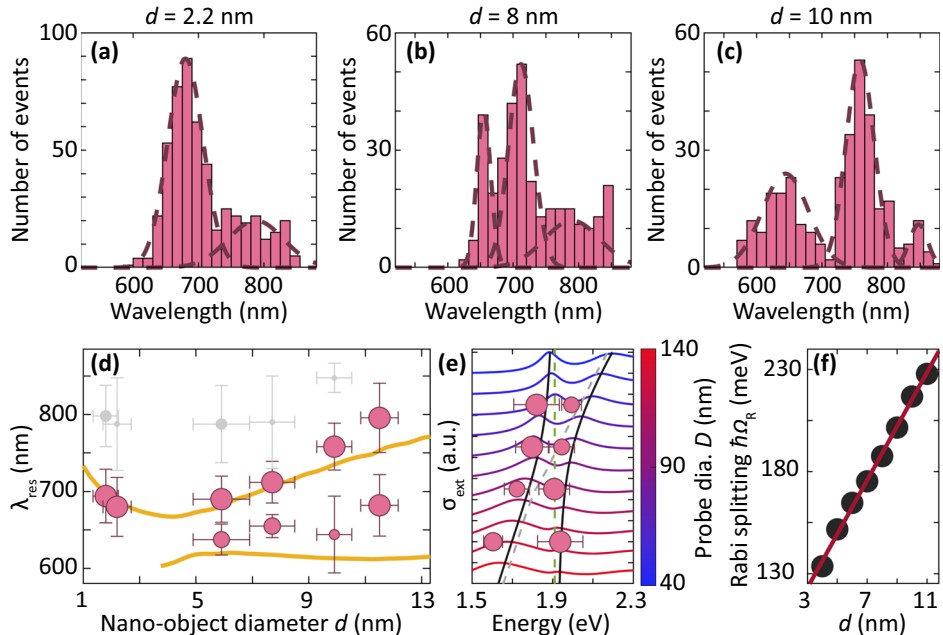

**Fig. 3 | Strong coupling of elastic light scattering between the nano-object and the cavity.** The histogram of the scattering peak of the measured nanocavities with a (**a**) 2.2 nm, (**b**) 8 nm, and (**c**) 10 nm nano-object inside the nanocavity. The dashed line is a Gaussian fit to the histogram. **d** The resonance wavelength of Au nano-object with a diameter of 1.8 nm to 12 nm inside a nanocavity measured experimentally (pink circles) and simulation result from Fig. 1c (yellow line). The size of the markers is determined by the relative frequency of occurrence in each histogram. The vertical error bar is determined from the standard deviation of the Gaussian distribution fit to the histograms, and the horizontal error bar is determined by the size variation of the nano-objects obtained from analyzing the SEM

images of the nano-objects. The gray markers are the scattering peaks that have a much lower occurrence frequency, and they are attributed to the presence of more than one nano-object inside the gap. **e** Experimentally measured resonance energy of an Au nano-object with a diameter of 6 nm inside a nanocavity for different nanoprobe diameters (pink circle). The blue-to-red colored lines are the simulated extinction cross-section spectra of a 6 nm nano-object inside a plasmonic nano-cavity. The black lines are based on a strongly coupled resonator model. The resonance of the nano-object on the Au film is the green dashed line, and that of the nanoprobe on the film is the gray dashed line. **f** Extracted Rabi splitting ($\Omega_R$) vs the nano-object diameter. The line shows a linear fit from an analytical model.

increased electron density inside the gap (for further simulations and analysis, see SI. I). We obtain Rabi splitting $\hbar\Omega_R$ as the energy separation between the coupled modes at the avoided crossing of the two resonance energies.

For a 6 nm nano-object in the nanocavity, the two modes consistently occur at different nanocavity resonances from numerical simulations (Fig. 3e). The energies of the modes follow a clear anti-crossing behavior, which fits well with the theoretical model (Eq. (1)). The extinction spectrum becomes broader at larger nanoprobe diameters due to increased damping by the larger nanoparticles[61]. Experimentally, we tune the nanocavity resonance using probe diameters between 60–125 nm. The experimental result also follows the strong coupling model in agreement with the simulation results. The Rabi splitting ($\hbar\Omega_R = 0.165$ eV) overcomes both the nanocavity loss ($\gamma_{nc} = 0.09$ eV) and nano-object on Au mirror loss ($\gamma_{no} = 0.04$ eV), indicating that the hybrid nano-object inside the nanocavity system is above the onset of the strong coupling regime (see SI. J for further analysis). We extract the Rabi splitting of different nano-object diameters from the numerical calculations. The Rabi splitting linearly increases from 130 meV to 230 meV for nano-object diameters 4 nm to 11 nm (Fig. 3f). The linear trend agrees with a simple model that we obtain for the Rabi splitting $\Omega_R = \frac{8}{3}\mu_m \frac{d}{D}\sqrt{\frac{2\pi c N_n}{3\hbar\lambda\varepsilon r_0}}$, where $\mu_m$ is the transition dipole moment of nano-object, $\hbar$ is Planck constant, $c$ is the light speed, $N_n$ is the metal atom density, $r_0$ is the atom radius and $\varepsilon$ is the permittivity of the medium inside the nanocavity gap, which includes the permittivity of the nano-object and air.

We extend our investigation to include nano-objects that are made of different metallic materials, to understand the effect of different dielectric functions of the nano-objects. We calculate the extinction cross-section of single gold, silver, aluminum, and copper

nano-objects inside the plasmonic nanocavity (SI. K). Two modes also appear in the extinction spectra for both nano-objects with different nanoprobe diameters, which follow the strong coupling model. We tune the resonance of the nanocavity for copper and silver nano-objects with a diameter of 6 nm (Fig. 4a). As the diameter of the nanoprobe increases, the relative peak value of the higher-energy mode decreases in comparison to that of the lower-energy mode. This is as a result of the larger nanoprobe size than the nano-object, which contributes more to the total extinction spectrum. We extract the Rabi splitting for gold, silver, aluminum, and copper nano-objects. There is a linear increase of the coupling strength to the plasma frequency ($\omega_p$) of the nano-object (Fig. 4b). This linear trend agrees well with an analytical model that describes the coupling strength as $\Omega_R = \omega_p \frac{\mu_m}{e}\sqrt{\frac{cmV_n}{\hbar\lambda\varepsilon V_m}}$ (SI. L), where $e$ is electron charge, $m$ is the effective electron mass, $V_m$ is the nanocavity mode volume and, $V_n$ is the nano-object volume.

## Scattering cross-section ratio

We now investigate how the nanocavity modifies the scattering cross-section of the nano-object, such that its signature is now observable in our measurements. In the weak coupling regime, far from the anti-crossing point, the longer wavelength mode tends to the resonance wavelength of the nano-object inside the cavity. Therefore, we compare the scattering cross-section of this longer wavelength mode to the resonance wavelength of the nano-object outside the nanocavity environment. We define $\eta_{sca}$ as the ratio of the scattering cross-section of the longer wavelength mode ($\sigma_{sca}^i$) to the scattering cross-section of nano-object mode outside of the nanocavity ($\sigma_{sca}^o$) i.e. $\eta_{sca} = \frac{\sigma_{sca}^i}{\sigma_{sca}^o}$, with the superscripts $i$ and $o$ indicating the $\sigma_{sca}$ inside and outside the

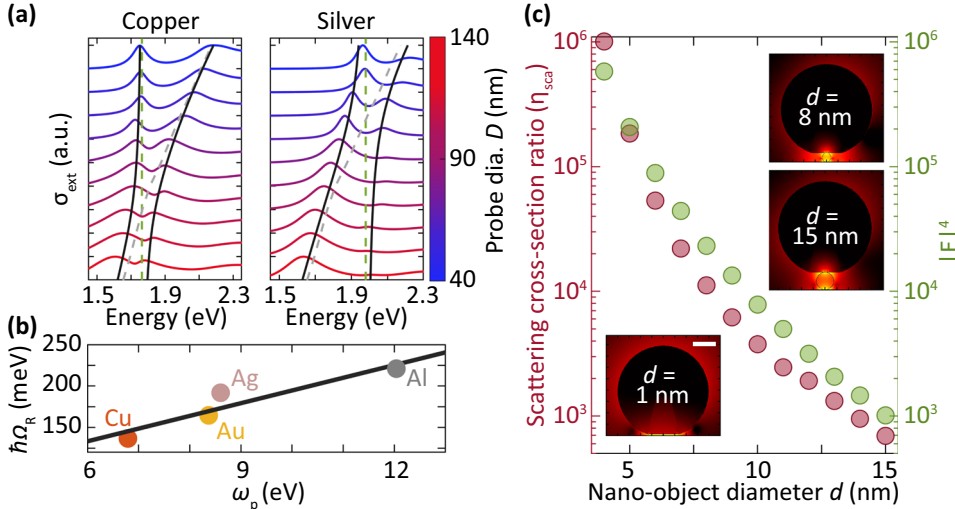

**Fig. 4 | Effect of the nano-object material property on strong coupling and enhanced scattering. a** Normalized extinction cross-section ($\sigma_{ext}$) spectra of a copper nano-object (left) and a silver nano-object (right) inside the nanocavity with different nanoprobe diameters. The black lines are based on the strongly coupled resonator model. The resonance of the nano-object on the Au film is the green dashed line, and that of the nanoprobe on the film is the gray dashed line. The diameter of the nano-object is 6 nm. **b** Rabi splitting of different nano-objects vs plasma frequency $\omega_p$. The line shows a linear fit from the analytical model. **c** Calculated scattering cross-section ratio of Au nano-object inside a nanocavity (red circle). The averaged electric field over the area of the nano-object $|E|^4$ in the gap of an empty nanocavity (green circle). The inset shows the electric field distribution of a nano-object in the nanocavity. The nanoprobe diameter is 80 nm, and the scale bar is 20 nm.

cavity, respectively. We consider the $\sigma_{sca}$ at about 100 nm (0.3 eV) away from the anti-crossing point i.e. in the weak coupling regime. The scattering ratio represents the efficiency of the nanocavity to boost the scattering cross-section of the nano-object, which is otherwise extremely challenging to detect in air.

There is a $\eta_{sca}$ of $10^6$ for a 4 nm nano-object, and it decreases to $10^3$ as the diameter of the nano-object increases to 15 nm (Fig. 4c). This implies that the nano-object exhibits a much larger scattering cross-section, thanks to the nanocavity. To understand the origin of the scattering ratio, we calculate the electric field enhancement distribution inside a nanocavity without the nano-object and average the electric field over the area of the nano-object inside the gap. Similar to surface-enhanced Raman scattering (SERS), the trend of the electric field $|E|^4$ vs $d$ follows closely that of the scattering cross-section ratio (Fig. 4c). The nanocavity localizes the electric field on the nano-object inside the gap (insets, Fig. 4c and Supplementary Fig. 15), resulting in a massive enhancement of the scattering cross-section ratio of the nano-object, as in SERS[70]. There is a deviation of less than one order of magnitude for larger nano-objects, and we attribute this deviation to the increased detuning of the nanocavity mode and nano-object on the Au film, which decreases the scattering cross-section ratio.

## Discussion

We have achieved elastic light scattering strong coupling between a sub-15 nm nano-object and a plasmonic nanocavity. Experimental and numerical results show a clear modification of the extinction spectra due to the presence of the nano-object in the cavity, revealing the signatures of nano-objects down to 1.8 nm. In addition, the extinction spectra reveal a clear anti-crossing behavior for nano-objects between 4 nm and 15 nm. Our results further show that the coupling strength is dependent on the plasma frequency of the nano-object in addition to their size. In comparison to free space, the nanocavity boosts the scattering cross-section by up to $10^6$, which enables the observation of the scattering signature of the nano-object. Moreover, this scattering cross-section ratio scales similarly as the electric field to the fourth power inside the plasmonic nanocavity, similar to SERS.

Our work establishes the first demonstration of strong coupling of elastic scattering of non-fluorescent, Raman inactive sub-15 nm

objects. The strong coupling effect provides a nonlinear wavelength dependence on the nano-object diameter, which results in a high sensitivity. Given the universal nature of elastic scattering, we anticipate applications in nanosensing and nanotechnology characterization of metallic nano-objects. Our approach holds promise for accurate imaging and detecting small nano-objects in a low-cost and easy-to-use manner. Due to the high sensitivity ($\frac{\partial \lambda_m}{\partial d}$) of up to 70 in the simulation and $35 \pm 9$ in the experiment, we foresee that future work will demonstrate the applicability to dielectric materials, which are in the weak coupling regime. Beyond the immediate application of detecting sub-15 nm nano-objects, our technique can be integrated into a wide range of photonic structures. For instance, interplay with bound states in the continuum could provide extremely sharp resonances with high quality factors, making detection even more selective and efficient. Moreover, the approach can be extended to hybrid systems exploiting Fano resonance modulations, enabling advanced control over scattering responses. The technique introduced here can complement electron microscopes and scanning probes for optical characterization. A nanoprobe attached to a cantilever can scan above the nano-object and the extinction spectrum can be collected through the cantilever.

## Methods

### Numerical setup

We solve the full Maxwell's equations using the boundary element method (BEM). The BEM is suitable for metallic nanostructures with abrupt interfaces that are embedded inside a homogeneous and isotropic dielectric medium. In addition, arbitrary shapes and sizes can be defined and computed with reasonable computational resources[53]. The approach is suitable for simulating metallic nanoparticles with sizes ranging from a few to a few hundred of nanometers[53,54]. The main advantage of the BEM approach over others (e.g. finite element method and finite difference time domain) is the speed of the computations, which enables us to explore a wide range of parameters.

We obtain the scattering cross-section and the extinction cross-section in the far-field and the electric field distribution in the vicinity of the nanostructures. In the simulations presented in this paper, the dielectric function of gold is taken from the Johnson and Christy

database[71] for nanocavity and the non-local model for nano-object. We use a perfectly spherical particle, described by the "trisphere" function in the MNPBEM MATLAB toolbox[53]. This function loads points on a sphere from a file and performs triangulation. It is known that the metallic nanoparticles are inevitably faceted. As a simplistic model that is relevant for experimental purposes, we simulate a faceted particle with a facet ratio $w/D = 0.375$, where $w$ and $D$ are facet size and the diameter of the nanoprobe. We choose this facet ratio based on previous measurements[64,72]. To realize the faceted-sphere structure, we cut the bottom side of a perfect sphere at $Z_{cut} = \sqrt{((D/2)^2 - w^2)}$ and then combine it with a 2D surface using the "polygon" function. We increase simulation accuracy by applying a fine triangular mesh with a maximum size of 2 nm to the nanoprobe facet and the region of the Au film where the nano-object interacts with the probe and film surface. The effect of non-locality of the nano-object is accounted for using the non-local hydrodynamic model introduced in the next section. To simulate the nano-object, we introduce a thin layer on the nano-object. The inner region of the nano-object, which separates the metal core from the artificial cover layer, is formed by displacing all vertices along the outward surface normal vectors. Special attention is required when refining the Green's function for neighboring boundary elements of the cover layer. Due to the thin thickness of the artificial layer, integration is done over two closely spaced boundary elements. We apply an 8-fold refined integration in polar coordinates relative to the nanoprobe, and subsequently pass the results to the BEM solver. This approach increases the number of integration points both for polar integration and for the integration over adjacent elements.

The structure shown in Fig. 1a is excited by a plane wave. The incident angle of the plane wave is 65 degrees due to the illumination angle in the dark-field scattering experiment. The polarization of the incident light is transverse magnetic with an amplitude of $E_0$ which is the unit amplitude in calculating electric field enhancement. The Au particle and the object are surrounded by air ($n = 1$) on top. The simulated wavelength range covered 482.5–960 nm, with a spacing of 2.5 nm (192 spectral points). We used a supercomputer (Snellius, Dutch National computer) with 48 nodes (Genoa), each node consisting of 192 cores and 1440 GB RAM capacity. To improve computational efficiency, the "parfor" construct in MATLAB is utilized to execute independent loop iterations in parallel. This method distributes the computational tasks across all available nodes, which leads to a substantial reduction in total execution time. Depending on the nanoprobe diameter and nano-object diameter, the computation time ranges from 9 min to 150 min per spectrum.

## Non-local hydrodynamic model

Surface plasmons on metal surfaces can concentrate light into sub-nanometric volumes. However, at scales near the Thomas-Fermi screening length, the electric response at the metal interface must be considered smeared out rather than localized[73]. We start with the polarizability of a spherical nano-object, including non-local effects, that is described by[74]:

$$\alpha_{no}^{NL} = 4\pi R^3 \left\{ \frac{\frac{\varepsilon_m}{\varepsilon_b} - \left[ 1 + \frac{\varepsilon_m - \varepsilon_b}{\varepsilon_b q_L R} \frac{i_1(q_L R)}{i_1'(q_L R)} \right]}{\frac{\varepsilon_m}{\varepsilon_b} + 2\left[ 1 + \frac{\varepsilon_m - \varepsilon_b}{\varepsilon_b q_L R} \frac{i_1(q_L R)}{i_1'(q_L R)} \right]} \right\}. \tag{2}$$

Here $q_L = \sqrt{\omega_P^2/\varepsilon_\infty - \omega(\omega + i\Gamma)}/\beta$ represent the longitudinal plasmon wavevector and $i_1$ is the modified spherical Bessel function of the first kind and its derivative, $\beta$ is proportional to the Fermi velocity, $\varepsilon_b$ is the dielectric function of the surrounding environment, and $R$ is the radius of the nano-object. The metal is characterized by a spatially dispersive permittivity tensor given by: $\varepsilon_m(\omega) = \varepsilon_\infty - \omega_P^2/[\omega(\omega + i\Gamma)]$, where $\varepsilon_\infty$ is

the dielectric background, $\Gamma$ and $\omega_P$ denote the metal damping and plasma frequency, respectively.

One can accurately reproduce the effects of non-locality using a model that replaces a non-local metallic surface with an effective local metallic surface coated with a dielectric layer[58]. The polarizability of a metallic nano-object with a thin dielectric layer of $\Delta d$ can be obtained from[58]:

$$\alpha_{no}^L = 4\pi R^3 \left\{ \frac{\frac{\varepsilon_m}{\varepsilon_b} - \frac{R}{R - \Delta d}\left(1 + \frac{\varepsilon_m}{\varepsilon_r^t}\frac{\Delta d}{R}\right)}{\frac{\varepsilon_m}{\varepsilon_b} + \frac{2R}{R - \Delta d}\left(1 + \frac{\varepsilon_m}{\varepsilon_r^t}\frac{\Delta d}{R}\right)} \right\}. \tag{3}$$

Comparing equations (3) and (2) and assuming that the dielectric layer is isotropic, the permittivity of the dielectric coating layer is given by[58]:

$$\varepsilon_t = \left( \frac{\varepsilon_b \varepsilon_m q_L \Delta d}{\varepsilon_m - \varepsilon_b} \right) \frac{i_1'(q_L R)}{i_1(q_L R)}. \tag{4}$$

Using a dielectric coating layer around the metallic nano-object with a diameter of 1 nm to 15 nm with the dielectric function introduced in the equation (4) was used to account for non-locality effects.

## Mesoscopic electromagnetic boundary condition

Utilizing mesoscopic boundary conditions is another approach that accounts for quantum mechanical mechanisms such as non-locality. It can correct the electromagnetic response and adjust the local classical simulation results that may deviate from experimental observations. We use a general theoretical framework based on Feibelman $d$-parameters to model the electromagnetic response of sub-15 nm nano-objects[59]. Yang et al. amend the classical electromagnetic boundary condition (Eq. (5)) through mesoscopic surface-response formalism by the Feibelman $d_\perp$ and $d_\parallel$ parameters (Eq. (6)).

$$
\begin{aligned}
D_\perp^+ - D_\perp^- &= 0 \\
B_\perp^+ - B_\perp^- &= 0 \\
\mathbf{E}_\parallel^+ - \mathbf{E}_\parallel^- &= \mathbf{0} \\
\mathbf{H}_\parallel^+ - \mathbf{H}_\parallel^- &= \mathbf{0}
\end{aligned}
\tag{5}
$$

$$
\begin{aligned}
D_\perp^+ - D_\perp^- &= d_\parallel \nabla_\parallel \cdot \left( \mathbf{D}_\parallel^+ - \mathbf{D}_\parallel^- \right) \\
B_\perp^+ - B_\perp^- &= 0 \\
\mathbf{E}_\parallel^+ - \mathbf{E}_\parallel^- &= -d_\perp \nabla_\parallel \left( E_\perp^+ - E_\perp^- \right) \\
\mathbf{H}_\parallel^+ - \mathbf{H}_\parallel^- &= i\omega d_\parallel \mathbf{D}_\parallel^+ - \mathbf{D}_\parallel^- \times \hat{\mathbf{n}},
\end{aligned}
\tag{6}
$$

which $\hat{\mathbf{n}}$ is unit normal vector and displacement $D$, magnetic $B$, electric $\mathbf{E}$ or magnetizing $\mathbf{H}$ field are oscillating at frequency $\omega$. The superscripts "+" and "−" indicate points just outside and inside the interface, and the subscripts "∥" and "⊥" denote parallel and perpendicular components. $d_\perp(\omega) = \frac{\int dx x \rho(x, \omega)}{\int dx \rho(x, \omega)}$ and $d_\parallel(\omega) = \frac{\int dx x \partial_x J_y(x, \omega)}{\int dx \partial_x J_y(x, \omega)}$, represent the frequency-dependent centroids of the induced charge $\rho(x, \omega)$ and the normal derivative of the tangential current $J_y(x, \omega)$[59]. The first Feibelman parameter, $d_\perp(\omega)$ is sufficient to describe most problems invoking metal surfaces and $d_\parallel(\omega) = 0$ on a metal-dielectric interface.

We use COMSOL Multiphysics to implement the mesoscopic boundary condition via the auxiliary potential method for sub-15 nm nano-objects. To improve the numerical stability, an integral form of weak constraints is used to reduce the localization errors. The results of the simulations using mesoscopic boundary conditions agree with the experimental results and non-local hydrodynamic model. However, the computational time using mesoscopic boundary conditions increased by about an order of magnitude in 2D while the non-local hydrodynamic simulations were performed in 3D.

## Sample fabrication

**Materials:** The Au nano-objects with diameters of $6 \pm 1$ nm, $8 + 1 - 0.5$ nm, $10 \pm 1$ nm, and $12 \pm 1$ nm were purchased from Aurion. The $1.8 \pm 0.45$ nm, $2.2 \pm 0.55$ nm, and 125 nm nanoparticles were obtained from Nanopartz. The 60 nm, 80 nm, and 100 nm nanoparticles, and cysteamine hydrochloride were purchased from the BBI solutions.

**Template-stripping:** We use the template-stripping technique to obtain the Au substrate. A silicon wafer was coated with a 100 nm layer of Au by thermal vapor deposition with a deposition rate of 0.3 Å/s. Then, the surface of the Au was coated with a thin layer of UV epoxy (Thorlabs, NOA81), and 8 mm round glasses were glued to the surface. The substrate was exposed to a UV light for 20 min. The glass substrates glued to the Au film on the initial wafer can be stored at room temperature in the air and peeled off as required (SI. N).

**Au nanoparticle functionalizing:** It is possible to use a similar surface modification technique for metals in general. Here, we choose gold nanoparticles for their inherent stability, resistance to oxidation under ambient conditions, and wide commercial availability in various sizes. In this work, we use 11 different Au nanoparticles with diameters from 1.8 nm to 125 nm.

The nanoprobes are functionalized with citrate, which is negatively charged due to the carboxylate group. The Au nano-objects are reactant-free and ready for functionalization. We use cysteamine to functionalize the surface of nano-objects. Cysteamine has a thiol group and an amino group, which can form a bond with the Au surface through the thiol group. The amino terminal group is positively charged, which attracts the negatively charged nanoprobe through electrostatic attraction. To start nano-object functionalization, the cysteamine hydrochloride is dissolved in Milli-Q water to create a 6.6 mM stock solution. We used a 1:1 volume ratio of cysteamine solution and colloidal Au nano-objects, and 1 mL of the nano-object colloidal solution with stock concentration was transferred to a microcentrifuge tube. 1 mL of the cysteamine hydrochloride solution with different concentrations (see Concentration of cysteamine) was mixed with Au nano-objects in the microcentrifuge tube to obtain cysteamine-functionalized Au nano-objects. The reaction mixture was centrifuged at 12000 rpm for 20 min, and then the supernatant was removed. The final solution is diluted for nanoparticle assembly in the next sample fabrication steps.

**Concentration of cysteamine:** Controlling the concentration of the cysteamine is crucial because excessive cysteamine on the surface of the Au nano-object results in a change in the resonance of the nano-object and can possibly cause aggregation of the nano-objects. We vary the cysteamine concentration from 1 $\mu$M to 20 $\mu$M, and the results indicate that 1 $\mu$M is the optimal concentration due to the minimal change in the optical properties of the nano-object (Supplementary Fig. 17). We use the absorbance ratio at 680 nm and 518 nm to determine the changes in the plasmon resonance of the nano-object.

**Au nanoparticle assembly:** To deposit the cysteamine-coated nano-object on the Au substrate, we dried the solution of the nano-objects on the Au substrate. We choose the surface density of 1600 particles/$\mu$m$^2$ for nano-object based on the nanoprobe facet size to avoid having more than 1 nano-object in the nanocavity but also provide sufficient nano-object on the surface. To assemble the nanoprobe on the nano-object, we drop cast the solution of the nanoprobe on the Au substrate with already fixed nano-objects. The deposition time of the nanoprobe determined the number of the formed nanocavities. We tuned the deposition time to ensure only one nanocavity is measured per time, and eventually, a 120-s deposition time was chosen. Finally, we rinsed the sample with deionized water to remove the unbonded nanoprobes.

**Dielectric layer-filled gap and empty gap nanocavities:** To obtain nanocavity with an empty gap, we dropcast the probe on the template-striped Au film for 5 min. We also fabricate nanocavities with a dielectric layer-filled gap by coating a 12 nm aluminum oxide ($Al_2O_3$) layer using electron-beam deposition. We use quartz crystal microbalance to measure the thickness of the coated layer, and a final thickness of 11.40 nm is obtained. Subsequently, nanoprobes are dropcasted on the $Al_2O_3$ coated Au film to form dielectric layer-filled gap nanocavities.

## Data availability

All data needed to evaluate the conclusions in the paper are present in the paper and/or the Supplementary Information. The data used in this study are available in https://doi.org/10.4121/6a23c9f4-d274-4a9b-aa07-8ee037ff0c85.

## Code availability

The codes needed to collect and analyze the data to evaluate the conclusions in the paper are available at https://github.com/MohammadRezaAghdaee/Spectroscopy.

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

## Acknowledgements

We acknowledge the Faculty of Science and Technology, the University of Twente, for a tenure-track start-up package. This publication is part of the project "Observing objects that evade light" with project number OCENW.XS23.1.198, which is financed by the Dutch Research Council. This work used the Dutch national e-infrastructure with the support of the SURF Cooperative using grant no. EINF-8959. We are thankful to Robert Molenaar for technical help and to Mike Dikkers for gold and aluminum oxide coating. We are also grateful to Ulrich Hohenester for the useful discussion, Herman Offerhaus for feedback, and Mireille Claessens for lab space and equipment.

## Author contributions

O.S.O. devised the project, the main conceptual ideas, and proof out-line. M.R.A. and O.S.O. constructed the experimental apparatus. M.R.A. prepared the samples, conducted the experiments, performed the analyses, and the numerical simulations, with help from O.S.O. M.J.G. carried out the SEM imaging and FIB milling. O.S.O. acquired funding and supervised the project. M.R.A. and O.S.O. wrote, reviewed, and edited the manuscript, and M.J.G. provided feedback.

## Competing interests

The authors declare no competing interests.
