## [Transparent Peer Review file · Nature Communications]

Optical detection of single sub-15 nm objects using elastic scattering strong coupling

Corresponding Author: Dr Oluwafemi Ojambati

Version 0:

Reviewer comments:

Reviewer #1

(Remarks to the Author)
Please find attached.

Reviewer #2

(Remarks to the Author)

The authors provided a manuscript relating to the detection of strong coupling in a nanoparticle system using dark-field micro-spectroscopy followed by numerical simulations. After reviewing the manuscript, the reviewer recommends the editor to either reject or to transfer the article to another journal for reasons that will be justified below.

1. The paper is heavily computational; given large amount of simulations and theory presented the reviewer believes that the work is more suited for a journal with an audience interested on the theoretical aspects of the work.
2. The authors deposited nanoparticles of various sizes onto gold film with an average spacing of 25 nm to ensure that on average only nano-object is under the probed facet (line 150-151). However, given the span of the particle spacing, the reviewer has concerns to whether the cavity only contains a single nano-object or a few (>1 but <10) or clusters of nano-objects. AFM is used to monitor the height of the large nano-particle (nano-probe) to distinguish the presence of the nano-object. However, given the low resolution AFM image provided and given that the nano-object is 10nm or less which is near the practical limits of the AFM, the review has concerns about the information extracted from the image due to correction issues.
3. Throughout the paper, the authors provided many graphs in the form of histogram plots with wavelength where the dashed lines are fitted to the observed events as a claim of theoretical verification. However some inconsistency can be observed in the manuscript. For example, in Figure 2g, two dashed curves are fitted for wavelengths greater than 600 nm where the events at shorter wavelengths are neglected. However, in Figure 2i the number of events at around 700 nm are fitted while neglect the regions <700 nm and between 700 and 800 nm despite their comparable amplitudes. Also, the peak fitting of curves in Figure 3a and 3b at 800 nm is not evident to the reviewer.
4. Figure 3d the authors provided a comparison between experimental data and their model. However, given the large scale error bars, the reviewer has doubts to whether the measured signal truly single particle and not just a few particles within the cavity.
5. Authors simulated the strong coupling effect in their system using eq(1) and citing reference 66. However, this model for strong coupling is typically reserved for systems of organic materials (dyes & molecules) and quantum dots that emit, to which the gold nano particle system explored in this work is not. Please justify the applicability of this model to the system.

6. Can the authors justify the effect of the nano-probe tilt on the nano particles? Since the nano-probes are deposited simply drop-casted on, there will be an effect on the equal spacing within the cavity.

7. Authors claim to a higher sensitivity is experimentally or theoretically shown? Not evident to the reviewer. For a claim of higher sensitivity, the authors need to prove experimentally and compare with a full survey standard techniques in the literature. The articles cited by the authors are simply a few special cases and not representative of the field as a whole.

Version 1:

Reviewer comments:

Reviewer #1

(Remarks to the Author)

Authors address the reviewer comments well.

(Remarks on code availability)

Authors address the reviewer comments well. I recommend accept.

Reviewer #2

(Remarks to the Author)

The authors have fully addressed the comments at a suitable level and included additional experimental and theoretical results to support the claims in the manuscript. I would like to recommend the editor to accept the revised manuscript for publication.

(Remarks on code availability)

REPLY TO REVIEWER 1

The article titled 'Optical detection of single sub-15 nm objects using elastic scattering strong coupling' by Aghdaee and Ojambati presents a very interesting report, very intriguing in the broad domain of plasmonics and nano-optics. The authors show experimental and numerical demonstration of elastic scattering strong coupling to detect single sub-15 nm nano-object. The nano-object strongly couples with a plasmonic nanocavity, which is formed by a Au nanoparticle placed on top a Au film. The strongly coupled system reveals two modes in darkfield extinction spectroscopy that strongly depend on the cavity and nano-object parameters.

The experiments and simulations are very well planned and executed, the results are very promising to the field of photonics. The results are super impressive and the work may be considered for acceptance following the major revision where authors should address the below mentioned comments.

1. The authors must clearly state which part of the dielectric constant modulation is actually supporting the detection of different types of nano-objects in the nanocavity.

We thank the reviewer for their comments.

Both the real and imaginary parts of the dielectric function play a role in detecting the different types of nano-objects inside the nanocavity. We have shown the dependence of the nano-object-in-cavity modes and Rabi splitting on the plasma frequency ω_p (page 9, second paragraph). For metals, the dielectric function is defined as $\varepsilon_m(\omega) = \varepsilon_\infty - \frac{\omega_p^2}{[\omega(\omega+i\Gamma)]'}$ where ω is the incident light frequency, Γ is the damping constant, and ε_∞ is the high-frequency permittivity limit. One can rewrite $\varepsilon_m(\omega)$ in two real and imaginary parts as $\varepsilon_m(\omega) = \varepsilon_\infty - \frac{\omega_p^2}{\omega^2 + \Gamma^2} + i \frac{\omega_p^2 \Gamma}{\omega(\omega^2 + \Gamma^2)}$. This equation clearly shows that both the real and imaginary parts of the dielectric function are functions of the plasma frequency. As we show in the paper, the Rabi splitting is strongly dependent on the plasma frequency (Fig. 4b). Therefore, changes in the real part or imaginary part of the dielectric function directly impact the Rabi splitting and nano-object-in-cavity modes, enabling the detection of different nano-objects. We have now revised the manuscript to clarify this in SI. L.

2. The experimental methods are not fully reported. The complex multi-architecture geometry -how are they exactly achieved. A diagrammatic representation of the substrate synthesis is essential as it is one of the main highlights of the work.

We have now provided a diagrammatic representation of the sample preparation process to clarify the fabrication of the nanocavity. Fig. R1, included in the revised manuscript (SI. N), outlines the step-by-step procedures for both Au film preparation and cavity assembly. The Au-coated silicon substrate is attached to a glass slide using epoxy and then UV-cured. Afterward, the Au layer can be peeled off to obtain a Au film. Nano-objects are functionalized with cysteamine, immobilized on the Au film surface, and subsequently assembled into the desired cavity structures through electrostatic attraction between the probe and the nano-objects. These additions provide a clear visualization of the fabrication process, enhancing the clarity of the experimental methodology.

Fig. R1: Sample fabrication process. (a) Template stripping process: Au film is deposited onto a silicon wafer; a UV-curable epoxy is applied to the Au surface; a glass coverslip is placed atop the epoxy; the sample is cured using UV light; and the coverslip is peeled off to retrieve the smooth Au surface. (b) nanocavity assembly process: functionalizing the nano-objects using cysteamine hydrochloride; immobilizing the nano-object on the surface by drying the solution of the nano-object on the Au film; drop casting functionalized nanoprobe, and forming a nanocavity, due to the electrostatic attraction of the nano-object and nanoprobe.

3. How about the TE and TM coupled modes. Generating a hybrid nano-assembly -does it yield coupling of both electric and magnetic flux modes in the hybrid infinitesimal nanogaps? Please discuss.

We agree with the reviewer that polarization plays a crucial role in mode excitation within the nanocavity. In particular, TM-polarized light, which has a significant electric field component along the z-direction, efficiently excites nano-object-in-cavity modes due to its strong interaction with the nanogap. The numerical results on polarization dependence, which we discuss in detail in SI. A reveals the hybrid nature of the nano-assembly, where both electric and magnetic fields couple in the confined nanogap. We have now emphasized the potential coupling of electric and magnetic field components in the nanogap in SI. A.

4. The concept of plasmonic hotspots is not given much of importance. Some very recent works from Cunningham group at UIUC may be considered to enrich the discussion on hottest hotspots and nano-assemblies.

Plasmonic hotspots are critical to the performance of our nanosystem, as they enable strong field confinement within the nanogap. We have now simulated and explored the electric field distribution of a full range of nano-object diameters (1-15 nm) and demonstrated how hotspot formation depends on object diameter. We now cite L. Leyang et al., Appl. Phys. Lett. 124.23 (2024) from the suggested group, as it is a good source that provides background information about hotspots. Following the reviewer's comment, we have added section SI. M to include the new simulation results and discussion on hotspots.

Fig. R2: Electric field distribution for nano-object inside the nanocavity with $D = 80$ nm and $d = 1$ nm to 15 nm.

5. How about the cost of this Custom-made dark-field micro-spectrometer? Please compare and contrast it and other functionalities with the ones that are in market or recent reports.

Our custom-made dark-field microspectrometer, designed for measuring scattering from individual nanocavities, costs around 40,000 €. Comparable commercial systems, such as the Horiba MicroHR, iHR320, and iHR550, typically range from 100,000 € to 150,000 €, making them 2.5 to 4 times more expensive. In terms of performance, our system provides a similar signal-to-noise ratio dark-field spectrum compared to existing reports (**1-** Hu et al., Nat. Commun. 15, 6835, 2024; **2-** Huang, Junyang, et al. Adv. Opt. Mater., 2025; **3-** S. Verlekar, et al. ACS nano 2025).

6. It is very important to introduce the readers to the concept of weak, moderate and strong coupling in the manuscript.

We have revised the main text (page 8, last paragraph) and SI. J to include a concise introduction to weak, moderate, and strong coupling. The relation between coupling strength and resonator losses defines coupling regimes. Weak coupling occurs when losses dominate, enhancing emission without spectral splitting. Moderate coupling occurs when the coupling strength becomes comparable to the losses, resulting in partial hybridization and spectral changes. Strong coupling occurs when the coupling strength exceeds the losses, resulting in clear Rabi splitting and hybridized light-matter states. This distinction emphasizes the strength of interaction in our work.

7. Some of the mathematical equations are not referenced. If they are not conventional ones, please provide the references to help other researchers in the broad domain of nano-optics and physics.

We apologize for the confusion. We have now referenced all nonconventional equations to support clarity.

8. The reviewer cannot see a high-resolution image of the sample that is made by the authors. We understand that a high end cross-sectioning would be necessary to view the assembly top-to-down. At least an AFM height profiling for layer 1, layer 1+1, and layer 1+1+1 is absolutely needed to support the wonderful claims made in this work.

We agree with the reviewer that high-resolution nanoscale imaging techniques would be a strong proof of the formation of the compound structure of nano-object-in-nanocavity system. First, we performed AFM height profiling of the Au film (layer 1), Au film with only nanoprobe (layer 1+1), and Au film with nano-object and nanoprobe on top (layer 1+1+1) (Sl. D), as the reviewer requested. In addition, we also performed a statistical analysis of several nanocavities with and without a nano-object. The results show that there is a height difference of about 12 nm (between layer 1+1 and layer 1+1+1), which is what we expect for the nano-object.

Futhermore, in line with the reviewer’s suggestion, we attempted focused ion beam (FIB) cross-sectioning for SEM imaging. Despite applying a protective carbon coating, the fragile nature of the compound nanostructure posed significant challenges. Accurate cross-sectioning requires precise targeting of the region directly beneath the nanoprobe, where the nano-object resides; however, our FIB system is constrained by a 100 nm step size in the y-direction (Fig. R3a). Consequently, most FIB cross-sections did not capture the nano-object within the gap under an 80 nm nanoprobe. In such cases, SEM imaging often revealed the nanoprobe at a height consistent with the diameter of the nano-object, supporting the presence of the nano-object beneath it (Fig. R3b,c). Despite these limitations, Fig. R3c shows an SEM image where the nanoprobe is assembled on top of the nano-object, thus further supporting the evidence of the formation of the compound structure.

Fig. R3 The SEM cross-sectioning. (a) The schematic of the SEM cross-sectional processes; (b,c) Brightfield image of a nanoprobe, the nano-objects on the sides are visible; (d) Two individual nano-objects sitting on the Au film on the left and a nanocavity with a nano-object inside the gap on the right. The scale bar is 10 nm.

9. The authors present very impressive results. However, the future scope and perspectives of this research are not mentioned well. Please add a section prior to conclusions mentioning numerous applications such novel platforms will be utilized in – especially photonic crystal, BIC, band edge

coupled modes, magnetic-electric flux coupling, Fano resonance modulations to name a few for human health and environmental health monitoring.

Beyond the immediate application of detecting sub-15 nm nano-objects, our technique can be integrated into a wide range of photonic structures. For instance, interplay with bound states in the continuum could provide extremely sharp resonances with high quality factors, making detection even more selective and efficient. Moreover, the approach can be extended to hybrid systems exploiting Fano resonance modulations, enabling advanced control over scattering responses. We have now discussed the future scope and potential applications of our platform, including suggestions from reviewers. We believe this addition strengthens the manuscript by providing a broader perspective on the impact of our research.

10. The details of simulations are not fully presented. The fine details required to reproduce the simulations in this work are not presented. Please include them comprehensively and draw effective correlations to the experimental results.

We have already discussed the simulation details in the Method section; however, considering the reviewer's comment, we have elaborated on the methods furthermore. We elaborate on the meshing of the nanostructure, which can affect the simulation results and the details of the hydrodynamic implementation on the system. We also discuss the details of optimizing the simulation time.

11. Please discuss the results from the perspective of metal-dielectric-metal interfaces, perhaps citing some of the early works by Economou. *Physical Review*, 182 (2) (1969), pp. 539-554. As in this work the fundamental idea is emerging by replacing the central dielectric with a nano-object indeed.

Economou (*Phys. Rev.* 182, 539–554, 1969) indeed provides the theoretical basis for understanding surface plasmons in metal-dielectric-metal (MIM) structures. Economou considered an idealized MIM system and derived the dispersion relation by applying boundary conditions at the two metal-dielectric interfaces. This dispersion relation is fundamental to understanding resonance conditions such as the Fabry-Perot modes, as further explored in C. Tserkezis et al., *Phys. Rev. A* 92, 053811, 2015.

In our study, the extinction cross-section spectra of empty nanocavities exhibit a dominant resonance, which we refer to as the nanocavity mode. This mode corresponds to the gap antenna mode (or I_1 mode) as described by Tserkezis et al. and originates from the confinement of electromagnetic fields in the plasmonic gap, which is conceptually related to the modes described by Economou. We observe nanocavity mode redshifts with increasing nanoprobe diameter, consistent with Tserkezis's report. Additionally, as the gap size increases, the nanocavity mode blueshifts, in agreement with theoretical predictions for MIM systems. Considering the reviewer's comments, we have now discussed the metal-dielectric-metal system in SI. B.

12. What are the merits and disadvantages of the developed platform as compared to the existing ones.

The strong coupling of elastic scattering from sub-15 nm non-fluorescent nano-objects enables highly sensitive, low-cost optical detection with broad applications in nanosensing and characterization. However, detecting dielectric, non-fluorescent nano-objects remains more challenging and requires further development. We have now elaborated more on the merits and disadvantages of the introduced platform in the discussion.

In general, the results presented in this research are of immense utility to the broad audience of nanotechnology, plasmonics, nano-optics, -both simulation and experimental experts. It may be eventually published, provided the authors address the above-mentioned comments satisfactorily.

REPLY TO REVIEWER 2

The authors provided a manuscript relating to the detection of strong coupling in a nanoparticle system using dark-field micro-spectroscopy followed by numerical simulations. After reviewing the manuscript, the reviewer recommends the editor to either reject or to transfer the article to another journal for reasons that will be justified below.

1. The paper is heavily computational; given large amount of simulations and theory presented the reviewer believes that the work is more suited for a journal with an audience interested on the theoretical aspects of the work.

We appreciate the reviewer's comment. We agree that our study includes analytical modeling and simulations to support and interpret the experimental findings, our major results. However, The primary focus of this work is the experimental demonstration of elastic scattering strong coupling between a single sub-15 nm nano-object and a plasmonic nanocavity. The theoretical and computational components are included to support the experimental observations rather than being the central focus. Moreover, approximately 70% of the figures in the main text present experimental data. In addition, we have provided additional data on scanning electron microscope cross-sectional imaging combined with focused ion beam milling of nano-object in nanocavity geometry (Fig. D5); scanning electron microscope imaging of the nano-objects, to determine the error margin of the nano-object sizes (D6); and AFM height profiling of many nano-object-in-nanocavity structures (Fig. D4). We note that these experimental data are challenging to acquire, as they push the limits of the state-of-the-art nanoscale imaging techniques.

2. The authors deposited nanoparticles of various sizes onto gold film with an average spacing of 25 nm to ensure that on average only nano-object is under the probed facet (line 150-151). However, given the span of the particle spacing, the reviewer has concerns to whether the cavity only contains a single nano-object or a few (>1 but <10) or clusters of nano-objects. AFM is used to monitor the height of the large nano-particle (nano-probe) to distinguish the presence of the nano-object. However, given the low resolution AFM image provided and given that the nano-object is 10nm or less which is near the practical limits of the AFM, the review has concerns about the information extracted from the image due to correction issues.

In the paper, we mentioned that it is statistically possible that two or more nano-objects might be inside the cavity. While the average spacing between nano-objects is 25 nm, there is a distribution of the nano-object spacing around the average value, as we extracted from the SEM image in Fig. 2b. To quantify the probability of having multiple nano-objects within a nanocavity, we performed a Monte Carlo simulation based on the experimentally derived distribution (Fig. 2b). Out of 100,000 trials simulating random nanoprobe placement, approximately 73% of nanocavities contained one nano-object, 25% contained two, and 2% had three (Fig. R4). This result indicates that while single occupancy is dominant, multiple-object scenarios are statistically probable. These results are consistent with our dark-field measurements, where the frequency of the peak attributed to 2 or more nano-objects in the cavity is 18% for a 2.2 nm nano-object, 22% for an 8 nm nano-object, and 19% for a 10 nm nano-object. In the dark-field measurement, we attribute the peak at 780 nm (Fig. 3a) to nanocavities containing multiple nano-objects. This interpretation is further confirmed by a control sample specifically fabricated with multiple nano-objects per cavity, which shows the same redshifted resonance peak (Fig. F8a).

To address the reviewer's concern regarding AFM resolution, we performed the AFM imaging with an AFM tip of 8 nm radius, which determines the lateral (xy) resolution. However, the relevant resolution is the AFM's z-resolution (< 0.1 nm) in the AFM height profiling to extract the height of the nano-object inside the nanocavities. The AFM's z-resolution is sufficient to resolve the height difference due to the nano-object (diameter = 12 nm) in the gap (Fig. 2c). We agree with the reviewer that the lateral

resolution of the AFM is low and thus, we resolve the fine lateral details (e.g. nano-object spacing on the Au film) using SEM imaging.

Fig. R4: Percentage distribution of the number of nano-objects present inside the nanocavity. The majority of cavities contain one nano-object (73%), followed by two nano-objects (25%), and a small fraction with three nano-objects (1.9%).

3. Throughout the paper, the authors provided many graphs in the form of histogram plots with wavelength where the dashed lines are fitted to the observed events as a claim of theoretical verification. However some inconsistency can be observed in the manuscript. For example, in Figure 2g, two dashed curves are fitted for wavelengths greater than 600 nm where the events at shorter wavelengths are neglected. However, in Figure 2i the number of events at around 700 nm are fitted while neglect the regions <700 nm and between 700 and 800 nm despite their comparable amplitudes. Also, the peak fitting of curves in Figure 3a and 3b at 800 nm is not evident to the reviewer.

The peaks at around 550 nm correspond to well-known dipolar mode of the nanoprobe, which often appear for different nanocavities, including the empty nanocavity (Fig. 2i) and nanocavity with a nano-object (Fig. 2g). This mode is often weak for TM excitation (Figs. 2d, 1c,d) and is not directly related to the nano-object-in-cavity coupling. Therefore dipolar mode of the nanoprobe is not included in the theoretical fitting, which focuses on nano-object-in-cavity modes. Regarding the weak resonance at <700 nm and between 700 and 800 nm (Fig. 2i), due to low occurrence, they were not originally included in the fitting; however, considering the reviewer's concern, we have now included the weak peak in Fig. 2i in the fitting.

The broad mode around 780 nm, seen in Figs. 2i, 3a, and 3b arises from nanocavities containing multiple nano-objects. Due to varying numbers of nano-objects and their position in the gap, these modes are broadened and less distinct, making individual peak fitting challenging. However, these events are included in the statistical analysis summarized in Figure 3d. We have now clarified this in the revised manuscript.

4. Figure 3d the authors provided a comparison between experimental data and their model. However, given the large scale error bars, the reviewer has doubts to whether the measured signal truly single particle and not just a few particles within the cavity.

The concern regarding the possibility of multiple particles within the cavity has been addressed in detail in our response to comment 2.

Regarding the horizontal error bars in Figure 3d, we have performed additional SEM measurements to determine the size distribution of the nanoparticles used, in contrast to using the specifications by the manufacturer. The SEM measurements confirm an uncertainty of, on average, less than 1 nm, which is consistent with the expected error for nanoparticles in the sub-15 nm size range (Fig. R5). For sub-15 nm nano-objects polydispersity index range of 20-25% is considered a common margin of error. Therefore, the horizontal error bars reflect the intrinsic variability of the particle size and do not

imply the presence of multiple particles. The measured signal remains consistent with single-particle events, as supported by both the experimental design and statistical analysis.

Fig. R5: Normalized histograms of nano-object diameter distributions obtained from SEM images for samples with nominal diameters of 1.8 nm, 6 nm, 8 nm, 10 nm, and 12 nm. Each histogram is color-coded by size group and fitted with a Gaussian function (solid lines) to extract the mean diameter and quantify size dispersion.

5. Authors simulated the strong coupling effect in their system using eq(1) and citing reference 66. However, this model for strong coupling is typically reserved for systems of organic materials (dyes & molecules) and quantum dots that emit, to which the gold nano particle system explored in this work is not. Please justify the applicability of this model to the system.

We agree with the reviewer that equation (1) is commonly applied to excitonic systems such as dyes, molecular emitters, and quantum dots. However, equation (1) is fundamentally derived from a general model describing the strong coupling between two harmonic oscillators and is not restricted to emissive systems. This formalism has also been applied in the literature to metallic nanoparticles and other non-emissive platforms (1- L. Yoon-Min, *Nano Converg.* 10.1 (2023): 34. 2-L. Novotny, *Am. J. Phys.* 78.11 (2010): 1199-1202. 3- S. Kotni, et al. *Nat. Commun.* 7.1 (2016): ncomms11823. 4- Dovzhenko, D. S., et al. *Nanoscale* 10.8 (2018): 3589-3605. 5- J. Heintz, et al. *ACS nano* 15.9 (2021): 14732-14743.) To address the concern, we have updated the reference to include a more general and broadly accepted source that validates the applicability of this model.

6. Can the authors justify the effect of the nano-probe tilt on the nano particles? Since the nano-probes are deposited simply drop-casted on, there will be an effect on the equal spacing within the cavity.

We thank the reviewer for this insightful comment. While we acknowledge that electrostatic interactions during drop-casting could potentially induce a tilt of the nanoprobe, cross-sectional SEM images (Fig. R3) confirm that the nanoprobe are positioned with a tilt angle close to 0° , supporting the assumption of uniform spacing within the cavity. To further assess the potential impact of probe tilt, we performed numerical simulations (see SI.G). These simulations indicate that a tilted probe leads to significant perturbations (redshift of up to 120 nm) of the cavity resonance modes. The strong agreement between the experimentally measured dark-field spectra and the simulation results for a 0° tilt provides additional evidence that the nanoprobe are effectively aligned within the cavity. Therefore, based on both experimental and computational results, we conclude that the effect of nanoprobe tilt on the equal spacing is negligible in our study.

Fig. R6 Extinction cross-section (σ_{ext}) spectra of a 7 nm Au nano-object inside a nanocavity (a) nanoprobe has 0° tilt, (b) nanoprobe has 32° tilt. The nanoprobe diameter is 80 nm.

7. Authors claim to a higher sensitivity is experimentally or theoretically shown? Not evident to the reviewer. For a claim of higher sensitivity, the authors need to prove experimentally and compare with a full survey standard techniques in the literature. The articles cited by the authors are simply a few special cases and not representative of the field as a whole.

We thank the reviewer for their feedback. As a proof of concept, we experimentally measured nanocavities with nano-objects of varying diameters, showing good agreement with simulation results (Fig. 3d). The higher sensitivity is based on a comparison with experimental techniques that detect nano-objects via shifts in resonance of nanophotonic structures. While only a limited number of studies have probed varying object sizes using this approach, we have now expanded the citations to include a broader range of representative works (Table S1.1).

The article titled ‘Optical detection of single sub-15 nm objects using elastic scattering strong coupling’ by Aghdaee and Ojambati presents a very interesting report, very intriguing in the broad domain of plasmonics and nano-optics. The authors show experimental and numerical demonstration of elastic scattering strong coupling to detect single sub-15 nm nano-object. The nano-object strongly couples with a plasmonic nanocavity, which is formed by a Au nanoparticle placed on top a Au film. The strongly coupled system reveals two modes in darkfield extinction spectroscopy that strongly depend on the cavity and nano-object parameters.

The experiments and simulations are very well planned and executed, the results are very promising to the field of photonics. The results are super impressive and the work may be considered for acceptance following the major revision where authors should address the below mentioned comments.

1. The authors must clearly state which part of the dielectric constant modulation is actually supporting the detection of different types of nano-objects in the nanocavity.
2. The experimental methods are not fully reported. The complex multi-architecture geometry -how are they exactly achieved. A diagrammatic representation of the substrate synthesis is essential as it is one of the main highlights of the work.
3. How about the TE and TM coupled modes. Generating a hybrid nano-assembly -does it yield coupling of both electric and magnetic flux modes in the hybrid infinitesimal nanogaps? Please discuss.
4. The concept of plasmonic hotspots is not given much of importance. Some very recent works from Cunningham group at UIUC may be considered to enrich the discussion on *hottest hotspots* and nano-assemblies.
5. How about the cost of this Custom-made dark-field micro-spectrometer? Please compare and contrast it and other functionalities with the ones that are in market or recent reports.
6. It is very important to introduce the readers to the concept of weak, moderate and strong coupling in the manuscript.
7. Some of the mathematical equations are not referenced. If they are not conventional ones, please provide the references to help other researchers in the broad domain of nano-optics and physics.
8. The reviewer cannot see a high-resolution image of the sample that is made by the authors. We understand that a high end cross-sectioning would be necessary to view the assembly top-to-down. At least an AFM height profiling for layer 1, layer 1+1, and layer 1+1+1 is absolutely needed to support the wonderful claims made in this work.
9. The authors present very impressive results. However, the future scope and perspectives of this research are not mentioned well. Please add a section prior to conclusions mentioning numerous applications such novel platforms will be utilized in – especially photonic crystal, BIC, band edge coupled modes, magnetic-electric flux coupling, Fano resonance modulations to name a few for human health and environmental health monitoring.
10. The details of simulations are not fully presented. The fine details required to reproduce the simulations in this work are not presented. Please include them comprehensively and draw effective correlations to the experimental results.

11. Please discuss the results from the perspective of metal-dielectric-metal interfaces, perhaps citing some of the early works by Economou. Physical Review, 182 (2) (1969), pp. 539-554. As in this work the fundamental idea is emerging by replacing the central dielectric with a nano-object in deed.
12. What are the merits and disadvantages of the developed platform as compared to the existing ones.

In general, the results presented in this research are of immense utility to the broad audience of nanotechnology, plasmonics, nano-optics, -both simulation and experimental experts. It may be eventually published, provided the authors address the above-mentioned comments satisfactorily.